# On a Theory of Nonparametric Pairwise Similarity for Clustering: Connecting Clustering to Classification

**Yingzhen Yang**[1] **Feng Liang**[1] **Shuicheng Yan**[2] **Zhangyang Wang**[1] **Thomas S. Huang**[1]

[1] University of Illinois at Urbana-Champaign, Urbana, IL 61801, USA
{yyang58,liangf,zwang119,t-huang1}@illinois.edu
[2] National University of Singapore, Singapore, 117576
eleyans@nus.edu.sg

## Abstract

Pairwise clustering methods partition the data space into clusters by the pairwise similarity between data points. The success of pairwise clustering largely depends on the pairwise similarity function defined over the data points, where kernel similarity is broadly used. In this paper, we present a novel pairwise clustering framework by bridging the gap between clustering and multi-class classification. This pairwise clustering framework learns an unsupervised nonparametric classifier from each data partition, and search for the optimal partition of the data by minimizing the generalization error of the learned classifiers associated with the data partitions. We consider two nonparametric classifiers in this framework, i.e. the nearest neighbor classifier and the plug-in classifier. Modeling the underlying data distribution by nonparametric kernel density estimation, the generalization error bounds for both unsupervised nonparametric classifiers are the sum of nonparametric pairwise similarity terms between the data points for the purpose of clustering. Under uniform distribution, the nonparametric similarity terms induced by both unsupervised classifiers exhibit a well known form of kernel similarity. We also prove that the generalization error bound for the unsupervised plug-in classifier is asymptotically equal to the weighted volume of cluster boundary [1] for Low Density Separation, a widely used criteria for semi-supervised learning and clustering. Based on the derived nonparametric pairwise similarity using the plug-in classifier, we propose a new nonparametric exemplar-based clustering method with enhanced discriminative capability, whose superiority is evidenced by the experimental results.

## 1 Introduction

Pairwise clustering methods partition the data into a set of self-similar clusters based on the pairwise similarity between the data points. Representative clustering methods include K-means [2] which minimizes the within-cluster dissimilarities, spectral clustering [3] which identifies clusters of more complex shapes lying on low dimensional manifolds, and the pairwise clustering method [4] using message-passing algorithm to inference the cluster labels in a pairwise undirected graphical model. Utilizing pairwise similarity, these pairwise clustering methods often avoid estimating complex hidden variables or parameters, which is difficult for high dimensional data.

However, most pairwise clustering methods assume that the pairwise similarity is given [2, 3], or they learn a more complicated similarity measure based on several given base similarities [4]. In this paper, we present a new framework for pairwise clustering where the pairwise similarity is derived as the generalization error bound for the unsupervised nonparametric classifier. The un-

supervised classifier is learned from unlabeled data and the hypothetical labeling. The quality of the hypothetical labeling is measured by the associated generalization error of the learned classifier, and the hypothetical labeling with minimum associated generalization error bound is preferred. We consider two nonparametric classifiers, i.e. the nearest neighbor classifier (NN) and the plug-in classifier (or the kernel density classifier). The generalization error bounds for both unsupervised classifiers are expressed as sum of pairwise terms between the data points, which can be interpreted as nonparametric pairwise similarity measure between the data points. Under uniform distribution, both nonparametric similarity measures exhibit a well known form of kernel similarity. We also prove that the generalization error bound for the unsupervised plug-in classifier is asymptotically equal to the weighted volume of cluster boundary [1] for Low Density Separation, a widely used criteria for semi-supervised learning and clustering.

Our work is closely related to discriminative clustering methods by unsupervised classification, which search for the cluster boundaries with the help of unsupervised classifier. For example, [5] learns a max-margin two-class classifier to group unlabeled data in an unsupervised manner, known as unsupervised SVM whose theoretical property is further analyzed in [6]. Also, [7] learns the kernel logistic regression classifier, and uses the entropy of the posterior distribution of the class label by the classifier to measure the quality of the learned classifier. More recent work presented in [8] learns an unsupervised classifier by maximizing the mutual information between cluster labels and the data, and the Squared-Loss Mutual Information is employed to produce a convex optimization problem. Although such discriminative methods produce satisfactory empirical results, the optimization of complex parameters hampers their application in high-dimensional data. Following the same principle of unsupervised classification using nonparametric classifiers, we derive nonparametric pairwise similarity and eliminate the need of estimating complicated parameters of the unsupervised classifer. As an application, we develop a new nonparametric exemplar-based clustering method with the derived nonparametric pairwise similarity induced by the plug-in classifier, and our new method demonstrates better empirical clustering results than the existing exemplar-based clustering methods.

It should be emphasized that our generalization bounds are essentially different from the literature. As nonparametric classification methods, the generalization properties of the nearest neighbor classifier (NN) and the plug-in classifier are extensively studied. Previous research focuses on the average generalization error of the NN [9, 10], which is the average error of the NN over all the random training data sets, or the excess risk of the plug-in classifier [11, 12]. In [9], it is shown that the average generalization error of the NN is bounded by twice of the Bayes error. Assuming that the class of the regression functions has a smooth parameter $\beta$, [11] proves that the excess risk of the plug-in classifier converges to 0 of the order $n^{\frac{-\beta}{2\beta+d}}$ where $d$ is the dimension of the data. [12] further shows that the plug-in classifier attains faster convergence rate of the excess risk, namely $n^{-\frac{1}{2}}$, under some margin assumption on the data distribution. All these generalization error bounds depend on the unknown Bayes error. By virtue of kernel density estimation and generalized kernel density estimation [13], our generalization bounds are represented mostly in terms of the data, leading to the pairwise similarities for clustering.

## 2 Formulation of Pairwise Clustering by Unsupervised Nonparametric Classification

The discriminative clustering literature [5, 7] has demonstrated the potential of multi-class classification for the clustering problem. Inspired by the natural connection between clustering and classification, we model the clustering problem as a multi-class classification problem: a classifier is learned from the training data built by a hypothetical labeling, which is a possible cluster labeling. The optimal hypothetical labeling is supposed to be the one such that its associated classifier has the minimum generalization error bound. To study the generalization bound for the classifier learned from the hypothetical labeling, we define the concept of classification model. Given unlabeled data $\{\mathbf{x}_l\}_{l=1}^n$, a classification model $M_{\mathcal{Y}}$ is constructed for any hypothetical labeling $\mathcal{Y} = \{\mathbf{y}_l\}_{l=1}^n$ as below:

**Definition 1.** *The classification model corresponding to the hypothetical labeling $\mathcal{Y} = \{\mathbf{y}_l\}_{l=1}^n$ is defined as $M_{\mathcal{Y}} = \left(\mathcal{S}, P_{XY}, \{\pi_i, f_i\}_{i=1}^Q, F\right)$. $\mathcal{S} = \{\mathbf{x}_l, \mathbf{y}_l\}_{l=1}^n$ are the labeled data by the*

*hypothetical labeling, and $\mathcal{S}$ are assumed to be i.i.d. samples drawn from the joint distribution $P_{XY} = P_{X|Y}P_Y$, where $(X, Y)$ is a random couple, $X \in \mathbb{R}^d$ represents the data and $Y \in \{1, 2, ..., Q\}$ is the class label of $X$, $Q$ is the number of classes determined by the hypothetical labeling. Furthermore, $P_{XY}$ is specified by $\{\pi^{(i)}, f^{(i)}\}_{i=1}^Q$ as follows: $\pi^{(i)}$ is the class prior for class i, i.e. $\Pr[Y = i] = \pi^{(i)}$; the conditional distribution $P_{X|Y=i}$ has probabilistic density function $f^{(i)}$, $i = 1, \ldots, Q$. $F$ is a classifier trained using the training data $\mathcal{S}$. The generalization error of the classification model $M_{\mathcal{Y}}$ is defined as the generalization error of the classifier $F$ in $M_{\mathcal{Y}}$.*

In this paper, we study two types of classification models with the nearest neighbor classifier and the plug-in classifier respectively, and derive their generalization error bounds as sum of pairwise similarity between the data. Given a specific type of classification model, the optimal hypothetical labeling corresponds to the classification model with minimum generalization error bound. The optimal hypothetical labeling also generates a data partition where the sum of pairwise similarity between the data from different clusters is minimized, which is a common criteria for discriminative clustering.

In the following text, we derive the generalization error bounds for the two types of classification models. Before that, we introduce more notations and assumptions for the classification model. Denote by $P_X$ the induced marginal distribution of $X$, and $f$ is the probabilistic density function of $P_X$ which is a mixture of $Q$ class-conditional densities: $f = \sum_{i=1}^Q \pi^{(i)} f^{(i)}$. $\eta^{(i)}(x)$ is the regression function of $Y$ on $X = x$, i.e. $\eta^{(i)}(x) = \Pr[Y = i | X = x] = \frac{\pi^{(i)} f^{(i)}(x)}{f(x)}$. For the sake of the consistency of the kernel density estimators used in the sequel, there are further assumptions on the marginal density and class-conditional densities in the classification model for any hypothetical labeling:

**(A)** $f$ is bounded from below, i.e. $f \geq f_{\min} > 0$

**(B)** $\{f^{(i)}\}$ is bounded from above, i.e. $f^{(i)} \leq f_{\max}^{(i)}$, and $f^{(i)} \in \Sigma_{\gamma, c_i}$, $1 \leq i \leq Q$.

where $\Sigma_{\gamma, c}$ is the class of Hölder-$\gamma$ smooth functions with Hölder constant $c$:

$$\Sigma_{\gamma, c} \triangleq \{f : \mathbb{R}^d \to \mathbb{R} \,|\, \forall x, y, |f(x) - f(y)| \leq c\|x - y\|^\gamma\}, \gamma > 0$$

It follows from assumption (B) that $f \in \Sigma_{\gamma, c}$ where $c = \sum_i \pi^{(i)} c_i$. Assumption (A) and (B) are mild. The upper bound for the density functions is widely required for the consistency of kernel density estimators [14, 15]; Hölder-$\gamma$ smoothness is required to bound the bias of such estimators, and it also appears in [12] for estimating the excess risk of the plug-in classifier. The lower bound for the marginal density is used to derive the consistency of the estimator of the regression function $\eta^{(i)}$ (Lemma 2) and the consistency of the generalized kernel density estimator (Lemma 3). We denote by $\mathcal{P}_X$ the collection of marginal distributions that satisfy assumption (A), and denote by $\mathcal{P}_{X|Y}$ the collection of class-conditional distributions that satisfy assumption (B). We then define the collection of joint distributions $\mathcal{P}_{XY}$ that $P_{XY}$ belongs to, which requires the marginal density and class-conditional densities satisfy assumption (A)-(B):

$$\mathcal{P}_{XY} \triangleq \{P_{XY} \mid P_X \in \mathcal{P}_X, \{P_{X|Y=i}\} \in \mathcal{P}_{X|Y}, \min_i \{\pi^{(i)}\} > 0\} \tag{1}$$

Given the joint distribution $P_{XY}$, the generalization error of the classifier $F$ learned from the training data $\mathcal{S}$ is:

$$R(F_{\mathcal{S}}) \triangleq \Pr[(X, Y) : F(X) \neq Y] \tag{2}$$

Nonparametric kernel density estimator (KDE) serves as the primary tool of estimating the underlying probabilistic density functions in our generalization analysis, and we introduce the KDE of $f$ as below:

$$\hat{f}_{n,h_n}(x) = \frac{1}{n} \sum_{l=1}^n K_{h_n}(x - \mathbf{x}_l) \tag{3}$$

where $K_h(x) = \frac{1}{h^d} K\left(\frac{x}{h}\right)$ is the isotropic Gaussian kernel with bandwidth $h$ and $K(x) \triangleq \frac{1}{(2\pi)^{d/2}} e^{-\frac{\|x\|^2}{2}}$. We have the following VC property of the Gaussian kernel $K$. Define the class

of functions

$$\mathcal{F} \triangleq \left\{ K\left(\frac{t-\cdot}{h}\right), t \in \mathbb{R}^d, h \neq 0 \right\} \tag{4}$$

The VC property appears in [14, 15, 16, 17, 18], and it is proved that $\mathcal{F}$ is a bounded VC class of measurable functions with respect to the envelope function $F$ such that $|u| \leq F$ for any $u \in \mathcal{F}$ (e.g. $F \equiv (2\pi)^{-\frac{d}{2}}$). It follows that there exist positive numbers $A$ and $v$ such that for every probability measure $P$ on $\mathbb{R}^d$ for which $\int F^2 dP < \infty$ and any $0 < \tau < 1$,

$$N\left(\mathcal{F}, \|\cdot\|_{L_2(P)}, \tau \|F\|_{L_2(P)}\right) \leq \left(\frac{A}{\tau}\right)^v \tag{5}$$

where $N\left(\mathcal{T}, \hat{d}, \epsilon\right)$ is defined as the minimal number of open $\hat{d}$-balls of radius $\epsilon$ required to cover $\mathcal{T}$ in the metric space $\left(\mathcal{T}, \hat{d}\right)$. $A$ and $v$ are called the VC characteristics of $\mathcal{F}$.

The VC property of $K$ is required for the consistency of kernel density estimators shown in Lemma 2. Also, we adopt the kernel estimator of $\eta^{(i)}$ below

$$\hat{\eta}_{n,h_n}^{(i)}(x) = \frac{\sum\limits_{l=1}^{n} K_{h_n}(x - \mathbf{x}_l) \mathbb{I}_{\{\mathbf{y}_l = i\}}}{n \hat{f}_{n,h_n}(x)} \tag{6}$$

Before stating Lemma 2, we introduce several frequently used quantities throughout this paper. Let $L, C > 0$ be constants which only depend on the VC characteristics of the Gaussian kernel $K$. We define

$$f_0 \triangleq \sum_{i=1}^{Q} \pi^{(i)} f_{\max}^{(i)} \quad \sigma_0^2 \triangleq \|K\|_2^2 f_0 \tag{7}$$

Also, for all positive numbers $\lambda \geq C$ and $\sigma > 0$, we define

$$E_{\sigma^2} \triangleq \frac{\log(1 + \lambda/4L)}{\lambda L \sigma^2} \tag{8}$$

Based on Corollary 2.2 in [14], Lemma 2 and Lemma 3 in the Appendix (more complete version in the supplementary) show the strong consistency (almost sure uniformly convergence) of several kernel density estimators, i.e. $\hat{f}_{n,h_n}$, $\{\hat{\eta}_{n,h_n}^{(i)}\}$ and the generalized kernel density estimator, and they form the basis for the derivation of the generalization error bounds for the two types of classification models.

## 3  Generalization Bounds

We derive the generalization error bounds for the two types of classification models with the nearest neighbor classifier and the plug-in classifier respectively. Substituting these kernel density estimators for the corresponding true density functions, Theorem 1 and Theorem 2 present the generalization error bounds for the classification models with the plug-in classifier and the nearest neighbor classifier. The dominant terms of both bounds are expressed as sum of pairwise similarity depending solely on the data, which facilitates the application of clustering. We also show the connection between the error bound for the plug-in classifier and Low Density Separation in this section. The detailed proofs are included in the supplementary.

### 3.1  Generalization Bound for the Classification Model with Plug-In Classifier

The plug-in classifier resembles the Bayes classifier while it uses the kernel density estimator of the regression function $\eta^{(i)}$ instead of the true $\eta^{(i)}$. It has the form

$$\text{PI}(X) = \underset{1 \leq i \leq Q}{\arg\max} \, \hat{\eta}_{n,h_n}^{(i)}(X) \tag{9}$$

where $\hat{\eta}_{n,h_n}^{(i)}$ is the nonparametric kernel estimator of the regression function $\eta^{(i)}$ by (6). The generalization capability of the plug-in classifier has been studied by the literature[11, 12]. Let

$F^*$ be the Bayes classifier, it is proved that the excess risk of $\text{PI}_S$, namely $\mathbb{E}_S R(\text{PI}_S) - R(F^*)$, converges to 0 of the order $n^{\frac{-\beta}{2\beta+d}}$ under some complexity assumption on the class of the regression functions with smooth parameter $\beta$ that $\{\eta^{(i)}\}$ belongs to [11, 12]. However, this result cannot be used to derive the generalization error bound for the plug-in classifier comprising of nonparametric pairwise similarities in our setting.

We show the upper bound for the generalization error of $\text{PI}_S$ in Lemma 1.

**Lemma 1.** *For any $P_{XY} \in \mathcal{P}_{XY}$, there exists a $n_0$ which depends on $\sigma_0$ and VC characteristics of $K$, when $n > n_0$, with probability greater than $1 - 2QLh_n^{E_{\sigma_0^2}}$, the generalization error of the plug-in classifier satisfies*

$$R(\text{PI}_S) \leq R_n^{\text{PI}} + \mathcal{O}\left(\sqrt{\frac{\log h_n^{-1}}{nh_n^d}} + h_n^\gamma\right) \tag{10}$$

$$R_n^{\text{PI}} = \sum_{i,j=1,\ldots,Q,i\neq j} \mathbb{E}_X\left[\hat{\eta}_{n,h_n}^{(i)}(X)\hat{\eta}_{n,h_n}^{(j)}(X)\right] \tag{11}$$

*where $E_{\sigma^2}$ is defined by (8), $h_n$ is chosen such that $h_n \to 0$, $\frac{\log h_n^{-1}}{nh_n^d} \to 0$, $\hat{\eta}_{n,h_n}^{(i)}$ is the kernel estimator of the regression function. Moreover, the equality in (10) holds when $\hat{\eta}_{n,h_n}^{(i)} \equiv \frac{1}{Q}$ for $1 \leq i \leq Q$.*

Based on Lemma 1, we can bound the error of the plug-in classifier from above by $R_n^{\text{PI}}$. Theorem 1 then gives the bound for the error of the plug-in classifier in the corresponding classification model using the generalized kernel density estimator in Lemma 3. The bound has a form of sum of pairwise similarity between the data from different classes.

**Theorem 1.** *(Error of the Plug-In Classifier) Given the classification model $\mathcal{M}_\mathcal{Y} = \left(\mathcal{S}, P_{XY}, \{\pi_i, f_i\}_{i=1}^Q, \text{PI}\right)$ such that $P_{XY} \in \mathcal{P}_{XY}$, there exists a $n_1$ which depends on $\sigma_0$, $\sigma_1$ and the VC characteristics of $K$, when $n > n_1$, with probability greater than $1 - 2QLh_n^{E_{\sigma_0^2}} - QLh_n^{E_{\sigma_1^2}}$, the generalization error of the plug-in classifier satisfies*

$$R(\text{PI}_S) \leq \hat{R}_n(\text{PI}_S) + \mathcal{O}\left(\sqrt{\frac{\log h_n^{-1}}{nh_n^d}} + h_n^\gamma\right) \tag{12}$$

*where $\hat{R}_n(\text{PI}_S) = \frac{1}{n^2}\sum_{l,m}\theta_{lm}G_{lm,\sqrt{2}h_n}$, $\sigma_1^2 = \frac{\|K\|_2^2 f_{\max}}{f_{\min}}$, $\theta_{lm} = \mathbb{1}_{\{\mathbf{y}_l \neq \mathbf{y}_m\}}$ is a class indicator function and*

$$G_{lm,h} = G_h(\mathbf{x}_l, \mathbf{x}_m), \quad G_h(x,y) = \frac{K_h(x-y)}{\hat{f}_{n,h}^{\frac{1}{2}}(x)\hat{f}_{n,h}^{\frac{1}{2}}(y)}, \tag{13}$$

*$E_{\sigma^2}$ is defined by (8), $h_n$ is chosen such that $h_n \to 0$, $\frac{\log h_n^{-1}}{nh_n^d} \to 0$, $\hat{f}_{n,h_n}$ is the kernel density estimator of $f$ defined by (3).*

$\hat{R}_n$ is the dominant term determined solely by the data and the excess error $\mathcal{O}\left(\sqrt{\frac{\log h_n^{-1}}{nh_n^d}} + h_n^\gamma\right)$ goes to 0 with infinite $n$. In the following subsection, we show the close connection between the error bound for the plug-in classifier and the weighted volume of cluster boundary, and the latter is proposed by [1] for Low Density Separation.

### 3.1.1 Connection to Low Density Separation

Low Density Separation [19], a well-known criteria for clustering, requires that the cluster boundary should pass through regions of low density. It has been extensively studied in unsupervised learning and semi-supervised learning [20, 21, 22]. Suppose the data $\{\mathbf{x}_l\}_{l=1}^n$ lies on a domain $\Omega \subseteq R^d$. Let $f$ be the probability density function on $\Omega$, $S$ be the cluster boundary which separates $\overline{\Omega}$ into two parts $S_1$ and $S_2$. Following the Low Density Separation assumption, [1] suggests that the

cluster boundary $S$ with low weighted volume $\int_S f(s)ds$ should be preferable. [1] also proves that a particular type of cut function converges to the weighted volume of $S$. Based on their study, we obtain the following result relating the error of the plug-in classifier to the weighted volume of the cluster boundary.

**Corollary 1.** *Under the assumption of Theorem 1, for any kernel bandwidth sequence $\{h_n\}_{n=1}^{\infty}$ such that $\lim\limits_{n\to\infty} h_n = 0$ and $h_n > n^{-\frac{1}{4d+4}}$, with probability 1,*

$$\lim_{n\to\infty} \frac{\sqrt{\pi}}{2h_n} \hat{R}_n\left(\mathrm{PI}_S\right) = \int_S f(s)ds \tag{14}$$

### 3.2 Generalization Bound for the Classification Model with Nearest Neighbor Classifier

Theorem 2 shows the generalization error bound for the classification model with nearest neighbor classifier (NN), which has a similar form as (12).

**Theorem 2.** *(Error of the NN) Given the classification model $M_{\mathcal{Y}} = \left(\mathcal{S}, P_{XY}, \{\pi_i, f_i\}_{i=1}^{Q}, \mathrm{NN}\right)$ such that $P_{XY} \in \mathcal{P}_{XY}$ and the support of $P_X$ is bounded by $[-M_0, M_0]^d$, there exists a $n_0$ which depends on $\sigma_0$ and VC characteristics of $K$, when $n > n_0$, with probability greater than $1 - 2QLh_n^{E_{\sigma_0^2}} - (2M_0)^d n^{dd_0} e^{-n^{1-dd_0} f_{\min}}$, the generalization error of the NN satisfies:*

$$R\left(\mathrm{NN}_S\right) \le \hat{R}_n\left(\mathrm{NN}_S\right) + c_0\left(\sqrt{d}\right)^{\gamma} n^{-d_0\gamma} + \mathcal{O}\left(\sqrt{\frac{\log h_n^{-1}}{nh_n^d}} + h_n^{\gamma}\right) \tag{15}$$

*where $\hat{R}_n\left(\mathrm{NN}\right) = \frac{1}{n} \sum\limits_{1\le l < m \le n} H_{lm,h_n}\theta_{lm}$,*

$$H_{lm,h_n} = K_{h_n}\left(\mathbf{x}_l - \mathbf{x}_m\right)\left(\frac{\int_{\mathcal{V}_l} \hat{f}_{n,h_n}(x)\,\mathrm{d}x}{\hat{f}_{n,h_n}(\mathbf{x}_l)} + \frac{\int_{\mathcal{V}_m} \hat{f}_{n,h_n}(x)\,\mathrm{d}x}{\hat{f}_{n,h_n}(\mathbf{x}_m)}\right), \tag{16}$$

*$E_{\sigma^2}$ is defined by (8), $d_0$ is a constant such that $dd_0 < 1$, $\hat{f}_{n,h_n}$ is the kernel density estimator of $f$ defined by (3) with the kernel bandwidth $h_n$ satisfying $h_n \to 0$, $\frac{\log h_n^{-1}}{nh_n^d} \to 0$, $\mathcal{V}_l$ is the Voronoi cell associated with $\mathbf{x}_l$, $c_0$ is a constant, $\theta_{lm} = \mathbb{I}_{\{\mathbf{y}_l \ne \mathbf{y}_m\}}$ is a class indicator function such that $\theta_{lm} = 1$ if $\mathbf{x}_l$ and $\mathbf{x}_m$ belongs to different classes, and $0$ otherwise. Moreover, the equality in (15) holds when $\eta^{(i)} \equiv \frac{1}{Q}$ for $1 \le i \le Q$.*

$G_{lm,\sqrt{2}h_n}$ in (13) and $H_{lm,h_n}$ in (16) are the new pairwise similarity functions induced by the plug-in classifier and the nearest neighbor classifier respectively. According to the proof of Theorem 1 and Theorem 2, the kernel density estimator $\hat{f}$ can be replaced by the true density $f$ in the denominators of (13) and (16), and the conclusions of Theorem 1 and 2 still hold. Therefore, both $G_{lm,\sqrt{2}h_n}$ and $H_{lm,h_n}$ are equal to ordinary Gaussian kernels (up to a scale) with different kernel bandwidth under uniform distribution, which explains the broadly used kernel similarity in data clustering from an angle of supervised learning.

## 4 Application to Exemplar-Based Clustering

We propose a nonparametric exemplar-based clustering algorithm using the derived nonparametric pairwise similarity by the plug-in classifier. In exemplar-based clustering, each $\mathbf{x}_l$ is associated with a cluster indicator $e_l$ ($l \in \{1, 2, ...n\}, e_l \in \{1, 2, ...n\}$), indicating that $\mathbf{x}_l$ takes $\mathbf{x}_{e_l}$ as the cluster exemplar. Data from the same cluster share the same cluster exemplar. We define $\mathbf{e} \triangleq \{e_l\}_{l=1}^{n}$. Moreover, a configuration of the cluster indicators $\mathbf{e}$ is consistent iff $e_l = l$ when $e_m = l$ for any $l, m \in 1..n$, meaning that $\mathbf{x}_l$ should take itself as its exemplar if any $\mathbf{x}_m$ take $\mathbf{x}_l$ as its exemplar. It is required that the cluster indicators $\mathbf{e}$ should always be consistent. Affinity Propagation (AP) [23], a representative of the exemplar-based clustering methods, solves the following optimization problem

$$\min_{\mathbf{e}} \sum_{l=1}^{n} S_{l,e_l} \quad s.t. \quad \mathbf{e} \text{ is consistent} \tag{17}$$

$S_{l,e_l}$ is the dissimilarity between $x_l$ and $x_{e_l}$, and note that $S_{l,l}$ is set to be nonzero to avoid the trivial minimizer of (17).

Now we aim to improve the discriminative capability of the exemplar-based clustering (17) using the nonparametric pairwise similarity derived by the unsupervised plug-in classifier. As mentioned before, the quality of the hypothetical labeling $\hat{\mathbf{y}}$ is evaluated by the generalization error bound for the nonparametric plug-in classifier trained by $S_{\hat{\mathbf{y}}}$, and the hypothetical labeling $\hat{\mathbf{y}}$ with minimum associated error bound is preferred, i.e. $\arg\min_{\hat{\mathbf{y}}} \hat{R}_n(\mathrm{PI}_S) = \arg\min_{\hat{\mathbf{y}}} \sum_{l,m} \theta_{lm} G_{lm,\sqrt{2}h_n}$ where $\theta_{lm} = \mathbb{I}_{\hat{y}_l \neq \hat{y}_m}$ and $G_{lm,\sqrt{2}h_n}$ is defined in (13). By Lemma 3, minimizing $\sum_{l,m} \theta_{lm} G_{lm,\sqrt{2}h_n}$ also enforces minimization of the weighted volume of cluster boundary asymptotically. To avoid the trivial clustering where all the data are grouped into a single cluster, we use the sum of within-cluster dissimilarities term $\sum_{l=1}^{n} \exp\left(-G_{le_l,\sqrt{2}h_n}\right)$ to control the size of clusters. Therefore, the objective function of our pairwise clustering method is below:

$$\Psi(e) = \sum_{l=1}^{n} \exp\left(-G_{le_l,\sqrt{2}h_n}\right) + \lambda \sum_{l,m}\left(\tilde{\theta}_{lm} G_{lm,\sqrt{2}h_n} + \rho_{lm}(e_l,e_m)\right) \tag{18}$$

where $\rho_{lm}$ is a function to enforce the consistency of the cluster indicators:

$$\rho_{lm}(e_l,e_m) = \begin{cases} \infty & e_m = l, e_l \neq l \text{ or } e_l = m, e_m \neq m \\ 0 & \text{otherwise} \end{cases},$$

$\lambda$ is a balancing parameter. Due to the form of (18), we construct a pairwise Markov Random Field (MRF) representing the unary term $u_l$ and the pairwise term $\tilde{\theta}_{lm} G_{lm,\sqrt{2}h_n} + \rho_{lm}$ as the data likelihood and prior respectively. The variables $\mathbf{e}$ are modeled as nodes and the unary term and pairwise term in (18) are modeled as potential functions in the pairwise MRF. The minimization of the objective function is then converted to a MAP (Maximum a Posterior) problem in the pairwise MRF. (18) is minimized by Max-Product Belief Propagation (BP).

The computational complexity of our clustering algorithm is $\mathcal{O}(TEN)$, where $E$ is the number of edges in the pairwise MRF, $T$ is the number of iterations of message passing in the BP algorithm. We call our new algorithm Plug-In Exemplar Clustering (PIEC), and compare it to representative exemplar-based clustering methods, i.e. AP and Convex Clustering with Exemplar-Based Model (CEB) [24], for clustering on three real data sets from UCI repository, i.e. Iris, Vertebral Column (VC) and Breast Tissue (BT). We record the average clustering accuracy (AC) and the standard deviation of AC for all the exemplar-based clustering methods when they produce the correct number of clusters for each data set with different values of $h_n$ and $\lambda$, and the results are shown in Table 1. Although AP produces better clustering accuracy on the VC data set, PIEC generates the correct cluster numbers for much more times. The dash in Table 1 indicates that the corresponding clustering method cannot produce the correct cluster number. The default value for the kernel bandwidth $h_n$ is $h_n^*$, which is set as the variance of the pairwise distance between data points $\{\|\mathbf{x}_l - \mathbf{x}_m\|_{l<m}\}$. The default value for the balancing parameter $\lambda$ is 1. We let $h_n = \alpha h_n^*$, $\lambda$ varies between $[0.2, 1]$ and $\alpha$ varies between $[0.2, 1.9]$ with step 0.2 and 0.05 respectively, resulting in 170 different parameter settings. We also generate the same number of parameter settings for AP and CEB.

Table 1: Comparison Between Exemplar-Based Clustering Methods. The number in the bracket is the number of times when the corresponding algorithm produces correct cluster numbers.

| Data sets | Iris | VC | BT |
|---|---|---|---|
| AP | $0.8933 \pm 0.0138$ (16) | **0.6677** (14) | 0.4906 (1) |
| CEB | $0.6929 \pm 0.0168$ (15) | $0.4748 \pm 0.0014$ (5) | $0.3868 \pm 0.08$ (2) |
| PIEC | **$0.9089 \pm 0.0033$** (15) | $0.5263 \pm 0.0173$ (35) | **$0.6585 \pm 0.0103$** (5) |

## 5   Conclusion

We propose a new pairwise clustering framework where nonparametric pairwise similarity is derived by minimizing the generalization error unsupervised nonparametric classifier. Our framework bridges the gap between clustering and multi-class classification, and explains the widely used kernel similarity for clustering. In addition, we prove that the generalization error bound for the unsupervised plug-in classifier is asymptotically equal to the weighted volume of cluster boundary for

Low Density Separation. Based on the derived nonparametric pairwise similarity using the plug-in classifier, we propose a new nonparametric exemplar-based clustering method with enhanced discriminative capability compared to the exiting exemplar-based clustering methods.

## Appendix

**Lemma 2.** *(Consistency of Kernel Density Estimator) Let the kernel bandwidth $h_n$ of the Gaussian kernel $K$ be chosen such that $h_n \to 0, \frac{\log h_n^{-1}}{nh_n^d} \to 0$. For any $P_X \in \mathcal{P}_X$, there exists a $n_0$ which depends on $\sigma_0$ and VC characteristics of $K$, when $n > n_0$, with probability greater than $1 - Lh_n^{E_{\sigma_0^2}}$ over the data $\{\mathbf{x}_l\}$,*

$$\left\| \hat{f}_{n,h_n}(x) - f(x) \right\|_\infty = \mathcal{O}\Big( \sqrt{\frac{\log h_n^{-1}}{nh_n^d}} + h_n^\gamma \Big) \tag{19}$$

*where $\hat{f}_{n,h_n}$ is the kernel density estimator of $f$. Furthermore, for any $P_{XY} \in \mathcal{P}_{XY}$, when $n > n_0$, then with probability greater than $1 - 2Lh_n^{E_{\sigma_0^2}}$ over the data $\{\mathbf{x}_l\}$,*

$$\left\| \hat{\eta}_{n,h_n}^{(i)}(x) - \eta^{(i)}(x) \right\|_\infty = \mathcal{O}\Big( \sqrt{\frac{\log h_n^{-1}}{nh_n^d}} + h_n^\gamma \Big) \tag{20}$$

*for each $1 \le i \le Q$.*

**Lemma 3.** *(Consistency of the Generalized Kernel Density Estimator) Suppose $f$ is the probabilistic density function of $P_X \in \mathcal{P}_X$. Let $g$ be a bounded function defined on $\mathcal{X}$ and $g \in \Sigma_{\gamma,g_0}, 0 < g_{\min} \le g \le g_{\max}$, and $e = \frac{f}{g}$. Define the generalized kernel density estimator of $e$ as*

$$\hat{e}_{n,h} \triangleq \frac{1}{n} \sum_{l=1}^n \frac{K_h(x - \mathbf{x}_l)}{g(\mathbf{x}_l)} \tag{21}$$

*Let $\sigma_g^2 = \frac{\|K\|_2^2 f_{\max}}{g_{\min}^2}$. There exists $n_g$ which depends on $\sigma_g$ and the VC characteristics of $K$, When $n > n_g$, with probability greater than $1 - Lh_n^{E_{\sigma_g^2}}$ over the data $\{\mathbf{x}_l\}$,*

$$\|\hat{e}_{n,h_n}(x) - e(x)\|_\infty = \mathcal{O}\Big( \sqrt{\frac{\log h_n^{-1}}{nh_n^d}} + h_n^\gamma \Big) \tag{22}$$

*where $h_n$ is chosen such that $h_n \to 0, \frac{\log h_n^{-1}}{nh_n^d} \to 0$.*

*Sketch of proof:* For fixed $h \ne 0$, we consider the class of functions

$$\mathcal{F}_g \triangleq \Big\{ \frac{K\left( \frac{t - \cdot}{h} \right)}{g(\cdot)}, t \in \mathbb{R}^d \Big\}$$

It can be verified that $\mathcal{F}_g$ is also a bounded VC class with the envelope function $F_g = \frac{F}{g_{\min}}$, and

$$N\left( \mathcal{F}_g, \|\cdot\|_{L_2(P)}, \tau \|F_g\|_{L_2(P)} \right) \le \left( \frac{A}{\tau} \right)^v \tag{23}$$

Then (22) follows from similar argument in the proof of Lemma 2 and Corollary 2.2 in [14]. □

The generalized kernel density estimator (21) is also used in [13] to estimate the Laplacian PDF Distance between two probabilistic density functions, and the authors only provide the proof of pointwise weak consistency of this estimator in [13]. Under mild conditions, our Lemma 3 and Lemma 2 show the strong consistency of the generalized kernel density estimator and the traditional kernel density estimator under the same theoretical framework of the VC property of the kernel.

**Acknowledgements.** This material is based upon work supported by the National Science Foundation under Grant No. 1318971.

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
