[Supplementary Material]

# Supplementary Document for On a Theory of Nonparametric Pairwise Similarity for Clustering: Connecting Clustering to Classification

**Yingzhen Yang**[1] **Feng Liang**[1] **Shuicheng Yan**[2] **Zhangyang Wang**[1] **Thomas S. Huang**[1]
[1] University of Illinois at Urbana-Champaign, Urbana, IL 61801, USA
{yyang58,liangf,zwang119,t-huang1}@illinois.edu
[2] National University of Singapore, Singapore, 117576
eleyans@nus.edu.sg

## 1    Proofs of Theorems and Lemmas in the Paper

We provide detailed proofs of the theorems and lemmas. As stated in the paper, we define

$$f_0 \triangleq \sum_{i=1}^{Q} \pi^{(i)} f_{\max}^{(i)} \quad \sigma_0^2 \triangleq \|K\|_2^2 f_0 \tag{1}$$

Let $L, C > 0$ be constants which only depend on the VC characteristics of the Gaussian kernel $K$. For all $\lambda \geq C$ and $\sigma > 0$, we define

$$E_{\sigma^2} \triangleq \frac{\log\left(1 + \lambda/4L\right)}{\lambda L \sigma^2} \tag{2}$$

**Lemma 1.** *For any $P_{XY} \in \mathcal{P}_{XY}$, there exists a $n_0$ which depends on $\sigma_0$ and VC characteristics of $K$, when $n > n_0$, with probability greater than $1 - 2QLh_n^{E_{\sigma_0^2}}$, the generalization error of the plug-in classifier satisfies*

$$R\left(\mathrm{PI}_S\right) \leq R_n^{\mathrm{PI}} + \mathcal{O}\left(\sqrt{\frac{\log h_n^{-1}}{nh_n^d}} + h_n^\gamma\right) \tag{3}$$

$$R_n^{\mathrm{PI}} = \sum_{i,j=1,\ldots,Q, i \neq j} \mathbb{E}_X\left[\hat{\eta}_{n,h_n}^{(i)}\left(X\right)\hat{\eta}_{n,h_n}^{(j)}\left(X\right)\right] \tag{4}$$

*where $E_{\sigma^2}$ is defined by (2), $h_n$ is chosen such that $h_n \to 0$, $\frac{\log h_n^{-1}}{nh_n^d} \to 0$, $\hat{\eta}_{n,h_n}^{(i)}$ is the kernel estimator of the regression function. Moreover, the equality in (3) holds when $\hat{\eta}_{n,h_n}^{(i)} \equiv \frac{1}{Q}$ for $1 \leq i \leq Q$.*

**Theorem 1.** *(Error of the Plug-In Classifier) Given the classification model $M_{\mathcal{Y}} = \left(\mathcal{S}, P_{XY}, \{\pi_i, f_i\}_{i=1}^{Q}, \mathrm{PI}\right)$ such that $P_{XY} \in \mathcal{P}_{XY}$, there exists a $n_1$ which depends on $\sigma_0$, $\sigma_1$ and the VC characteristics of $K$, when $n > n_1$, with probability greater than $1 - 2QLh_n^{E_{\sigma_0^2}} - QLh_n^{E_{\sigma_1^2}}$, the generalization error of the plug-in classifier satisfies*

$$R\left(\mathrm{PI}_S\right) \leq \hat{R}_n\left(\mathrm{PI}_S\right) + \mathcal{O}\left(\sqrt{\frac{\log h_n^{-1}}{nh_n^d}} + h_n^\gamma\right) \tag{5}$$

*where $\hat{R}_n\left(\mathrm{PI}_S\right) = \frac{1}{n^2}\sum_{l,m}\theta_{lm}G_{lm,\sqrt{2}h_n}$, $\sigma_1^2 = \frac{\|K\|_2^2 f_{\max}}{f_{\min}}$, $\theta_{lm} = \mathbb{1}_{\{\mathbf{y}_l \neq \mathbf{y}_m\}}$ is a class indicator function and*

$$G_{lm,h} = G_h\left(\mathbf{x}_l, \mathbf{x}_m\right), \ G_h\left(x, y\right) = \frac{K_h\left(x - y\right)}{\hat{f}_{n,h}^{\frac{1}{2}}\left(x\right)\hat{f}_{n,h}^{\frac{1}{2}}\left(y\right)}, \

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

$\varlimsup_{n \to \infty} \|\hat{e}_{n,h_n} \left( x \right) - e \left( x \right)\|_\infty = \mathcal{O}\big( \sqrt{\frac{\log h_n^{-1}}{n h_n^d}} + h_n^\gamma \big).$

## 1.1 Proof of Lemma 3

*Proof.* Since $f$ satisfies assumption (A), applying Corollary 2.2 in [1], for $L, C > 0$ that depend solely on the VC characteristics of $K$ and any $\lambda > C$, when $n > n_0$,

$$\Pr \left[ \left\| \hat{f}_{n,h_n} - \mathbb{E} \left[ \hat{f}_{n,h_n} \right] \right\|_\infty \geq \tau_n \right] \leq L \exp \left( -\frac{1}{L} \frac{\log \left( 1 + \lambda/4L \right)}{\lambda} \frac{n h_n^d \tau_n^2}{\sigma_0^2} \right) \tag{a}$$

where $\tau_n = g \sqrt{\frac{\log h_n^{-1}}{n h_n^d}}$, $g$ is a constant and $g > 1$.

Also,

$$\left\| \mathbb{E} \left[ \hat{f}_{n,h_n} \right] - f \left( x \right) \right\|_\infty = \| \mathbb{E}_Z \left[ K_h \left( x - Z \right) \right] - f \left( x \right) \|_\infty \tag{b}$$

$$= \left\| \int_{\mathcal{X}} f \left( x - h_n z \right) K \left( z \right) dz - f \left( x \right) \int_{\mathcal{X}} K \left( z \right) dz \right\|_\infty$$

$$\leq \int_{\mathcal{X}} \| f \left( x - h_n z \right) - f \left( x \right) \|_\infty K \left( z \right) dz$$

$$\leq c h_n^\gamma \int_{\mathcal{X}} \| z \|^\gamma K \left( z \right) dx = c h_n^\gamma K_\gamma$$

because $f$ is a Hölder-$\gamma$ smooth function with Hölder constant $c = \sum_i \pi^{(i)} c_i$, and $\mathcal{X} = \mathbb{R}^d$. Based on $(a)$ and $(b)$, with probability greater than $1 - L h_n^{E_{\sigma_0^2}}$ (since $h_n^{g^2 E_{\sigma_0^2}} < h_n^{E_{\sigma_0^2}}$ when $h_n < 1$ for sufficiently large $n$) over the data $\{\mathbf{x}_l\}$, (10) holds.

Moreover,

$$\left\| \hat{\eta}_{n,h_n}^{(i)} \left( x \right) - \eta^{(i)} \left( x \right) \right\|_\infty \leq \left\| \hat{\eta}_{n,h_n}^{(i)} \left( x \right) - \frac{\sum_{l=1}^{n} K_{h_n} \left( x - \mathbf{x}_l \right) \mathbb{I}_{\{\mathbf{y}_l = i\}}}{n f \left( x \right)} \right\|_\infty + \left\| \frac{\sum_{l=1}^{n} K_{h_n} \left( x - \mathbf{x}_l \right) \mathbb{I}_{\{\mathbf{y}_l = i\}}}{n f \left( x \right)} - \eta^{(i)} \left( x \right) \right\|_\infty$$

$$\leq \left\| \frac{\sum_{l=1}^{n} K_{h_n} \left( x - \mathbf{x}_l \right) \mathbb{I}_{\{\mathbf{y}_l = i\}}}{n} \frac{f \left( x \right) - \hat{f}_{n,h_n} \left( x \right)}{f \left( x \right) \hat{f}_{n,h_n} \left( x \right)} \right\|_\infty + \left\| \frac{1}{f \left( x \right)} \left( \frac{\sum_{l=1}^{n} K_{h_n} \left( x - \mathbf{x}_l \right) \mathbb{I}_{\{\mathbf{y}_l = i\}}}{n} - \pi^{(i)} f^{(i)} \left( x \right) \right) \right\|_\infty$$

$$\leq \frac{1}{f_{\min}} \left\| f \left( x \right) - \hat{f}_{n,h_n} \left( x \right) \right\|_\infty + \frac{1}{f_{\min}} \left\| \frac{\sum_{l=1}^{n} K_{h_n} \left( x - \mathbf{x}_l \right) \mathbb{I}_{\{\mathbf{y}_l = i\}}}{n} - \pi^{(i)} f^{(i)} \left( x \right) \right\|_\infty$$

Similar to the proof of (10), with probability greater than $1 - L h_n^{E_{\sigma_0^2}}$ over the data $\{\mathbf{x}_l\}$, $\left\| \frac{\sum_{l=1}^{n} K_{h_n} \left( x - \mathbf{x}_l \right) \mathbb{I}_{\{\mathbf{y}_l = i\}}}{n} - \pi^{(i)} f^{(i)} \left( x \right) \right\|_\infty = \mathcal{O} \left( \sqrt{\frac{\log h_n^{-1}}{n h_n^d}} + h_n^\gamma \right)$. Also, with probability greater

than $1 - Lh_n^{E_{\sigma_0^2}}$, $\left\| \hat{f}_{n,h_n}(x) - f(x) \right\|_\infty = \mathcal{O}\left( \sqrt{\frac{\log h_n^{-1}}{nh_n^d}} + h_n^\gamma \right)$. Therefore, with probability greater than $1 - 2Lh_n^{E_{\sigma_0^2}}$, (12) holds. (11) and (13) follow from an application of the Borel-Cantelli lemma. $\qquad\square$

## 1.2 Proof of Lemma 3

*Proof.* For fixed $h \neq 0$, we consider the class of functions

$$\mathcal{F} \triangleq \{K\left(\frac{t-\cdot}{h}\right), t \in \mathbb{R}^d\} \quad \mathcal{F}_g \triangleq \{\frac{K\left(\frac{t-\cdot}{h}\right)}{g(\cdot)}, t \in \mathbb{R}^d\}$$

Since $\mathcal{F}$ is a bounded VC class of measurable functions, there exist positive numbers $A$ and $v$ such that for every probability measure $P$ on $\mathbb{R}^d$ for which $\int F^2 dP < \infty$ and any $0 < \tau < 1$,

$$N\left(\mathcal{F}, \|\cdot\|_{L_2(P)}, \tau \|F\|_{L_2(P)}\right) \leq \left(\frac{A}{\tau}\right)^v \tag{16}$$

For any $t_1$ and $t_2$, $\left\| \frac{K\left(\frac{t_1-\cdot}{h}\right)}{g(\cdot)} - \frac{K\left(\frac{t_2-\cdot}{h}\right)}{g(\cdot)} \right\|_{L_2(P)} \leq \frac{1}{g_{\min}} \left\| K\left(\frac{t_1-\cdot}{h}\right) - K\left(\frac{t_2-\cdot}{h}\right) \right\|_{L_2(P)}$. Let $B_\mathcal{F}(t_0, \delta) \triangleq \{t : \left\| K\left(\frac{t-\cdot}{h}\right) - K\left(\frac{t_0-\cdot}{h}\right) \right\|_{L_2(P)} \leq \delta\}$, and $B_{\mathcal{F}_g}(t_0, \delta) \triangleq \{t : \left\| \frac{K\left(\frac{t-\cdot}{h}\right)}{g(\cdot)} - \frac{K\left(\frac{t_0-\cdot}{h}\right)}{g(\cdot)} \right\|_{L_2(P)} \leq \delta\}$. Then $B_\mathcal{F}(t_0, \delta) \subseteq B_{\mathcal{F}_g}\left(t_0, \frac{\delta}{g_{\min}}\right)$.

We choose the envelope function for $\mathcal{F}_g$ as $F_g = \frac{F}{g_{\min}}$ and $|u_g| \leq F_g$ for any $u_g \in \mathcal{F}_g$, then

$$N\left(\mathcal{F}_g, \|\cdot\|_{L_2(P)}, \tau \|F_g\|_{L_2(P)}\right) \leq \left(\frac{A}{\tau}\right)^v \tag{17}$$

So that $\mathcal{F}_g$ is also a bounded VC class. The conclusion follows from similar argument in the proof of Theorem 1 and Corollary 2.2 in [1]. $\qquad\square$

## 1.3 Proof of Lemma 1

*Proof.* Let $P_{XY} \in \mathcal{P}_{XY}$. It can be verified that

$$R(\text{PI}_S) = \sum_{i,j=1,\ldots,Q, i \neq j} \mathbb{E}_X \left[ \eta^{(i)}(X) \Pr[\text{PI}_S(X) = j] \right] \tag{18}$$

According to Lemma 2 and (18), with probability greater than $1 - 2QLh_n^{E_{\sigma_0^2}}$,

$$R(\text{PI}_S) = \sum_{i \neq j} \mathbb{E}_X \left[ \hat{\eta}_{n,h_n}^{(i)}(X) \Pr[\text{PI}_S(X) = j] \right] + \mathcal{O}\left( \sqrt{\frac{\log h_n^{-1}}{nh_n^d}} + h_n^\gamma \right)$$

Denote by $\{\mathbf{R}_1, \mathbf{R}_2, \ldots \mathbf{R}_Q\}$ the decision regions of $\text{PI}_S$, then $\hat{\eta}_{n,h_n}^{(i)} \geq \hat{\eta}_{n,h_n}^{(i')}$ for all $i' \neq i$ on each $\mathbf{R}_i$, and

$$\sum_{i,j=1,\ldots,Q, i \neq j} \mathbb{E}_X \left[ \hat{\eta}_{n,h_n}^{(i)}(X) \Pr[\text{PI}_S(X) = j] \right]$$

$$= \sum_{i,j=1,\ldots,Q, i \neq j} \mathbb{E}_{X \in \mathbf{R}_j} \left[ \hat{\eta}_{n,h_n}^{(i)}(X) \cdot \sum_{k=1}^Q \hat{\eta}_{n,h_n}^{(k)}(X) \right]$$

$$\leq \mathbb{E}_X \left[ \left( \sum_{k=1}^Q \hat{\eta}_{n,h_n}^{(k)}(X) \right)^2 \right] - \sum_{i=1}^Q \mathbb{E}_X \left[ \left( \hat{\eta}_{n,h_n}^{(i)}(X) \right)^2 \right]$$

$$= \sum_{i,j=1,\ldots,Q, i \neq j} \mathbb{E}_X \left[ \hat{\eta}_{n,h_n}^{(i)}(X) \hat{\eta}_{n,h_n}^{(j)}(X) \right] \tag{19}$$

Therefore we obtain (3), and the equality in (3) holds when $\hat{\eta}_{n,h_n}^{(i)} \equiv \frac{1}{Q}$ for $1 \leq i \leq Q$. $\qquad\square$

## 1.4 Proof of Theorem 1

*Proof.* By Lemma 2 and Lemma 1, there exists an $n^{(1)}$ which depends on $\sigma_0$ and the VC characteristics of $K$, when $n > n^{(1)}$, with probability greater than $1 - 2QLh_n^{E_{\sigma_0^2}}$,

$$R_n^{\text{PI}} = \sum_{i \neq j} \mathbb{E}_X \left[ \hat{\eta}_{n,h_n}^{(i)}(X) \, \hat{\eta}_{n,h_n}^{(j)}(X) \right]$$

$$= \sum_{i \neq j} \mathbb{E}_X \left[ \eta^{(i)}(X) \, \eta^{(j)}(X) \right] + \mathcal{O}\left( \sqrt{\frac{\log h_n^{-1}}{nh_n^d}} + h_n^\gamma \right) \tag{20}$$

where $h_n \to 0$, $\frac{\log h_n^{-1}}{nh_n^d} \to 0$. Note that

$$\mathbb{E}_X \left[ \eta^{(i)}(X) \, \eta^{(j)}(X) \right] = \int_{\mathcal{X}} \frac{\pi^{(i)} f^{(i)}(x)}{f^{\frac{1}{2}}(x)} \cdot \frac{\pi^{(j)} f^{(j)}(x)}{f^{\frac{1}{2}}(x)} \, dx,$$

Using the generalized kernel density estimator (14), we obtain the kernel estimator $\tilde{\eta}_{n,h_n}^{(i)}$ of $\frac{\pi^{(i)} f^{(i)}(x)}{f^{\frac{1}{2}}(x)}$ as below:

$$\tilde{\eta}_{n,h_n}^{(i)}(x) = \frac{1}{n} \sum_{l=1}^{n} \frac{K_{h_n}(x - \mathbf{x}_l) \, \mathbb{I}_{\{\mathbf{y}_l = i\}}}{f^{\frac{1}{2}}(\mathbf{x}_l)} \tag{21}$$

By Lemma 3, there exists an $n^{(2)}$ which depends on $\sigma_1$ and the VC characteristics of $K$, when $n > n^{(2)}$, with probability greater than $1 - QLh_n^{E_{\sigma_1^2}}$,

$$\sum_{i \neq j} \mathbb{E}_X \left[ \eta^{(i)}(X) \, \eta^{(j)}(X) \right] = \sum_{i \neq j} \mathbb{E}_X \left[ \tilde{\eta}_{n,h_n}^{(i)}(X) \, \tilde{\eta}_{n,h_n}^{(j)}(X) \right] + \mathcal{O}\left( \sqrt{\frac{\log h_n^{-1}}{nh_n^d}} + h_n^\gamma \right) \tag{22}$$

By convolution theorem of Gaussian kernels,

$$\sum_{i \neq j} \mathbb{E}_X \left[ \tilde{\eta}_{n,h_n}^{(i)}(X) \, \tilde{\eta}_{n,h_n}^{(j)}(X) \right] = \frac{1}{n^2} \sum_{l,m} \frac{K_{\sqrt{2}h_n}(\mathbf{x}_l - \mathbf{x}_m)}{f^{\frac{1}{2}}(\mathbf{x}_l) f^{\frac{1}{2}}(\mathbf{x}_m)} \theta_{lm}$$

Let $\tilde{h}_n = \sqrt{2}h_n$, there exists $n^{(3)}$ depending on $\sigma_0$ and the VC characteristics of $K$, when $n > n^{(3)}$, with probability greater than $1 - Lh_n^{E_{\sigma_0^2}}$, $\|\hat{f}_{n,\tilde{h}_n}(x) - f(x)\|_\infty = \mathcal{O}\left( \sqrt{\frac{\log \tilde{h}_n^{-1}}{n\tilde{h}_n^d}} + \tilde{h}_n^\gamma \right)$ and $\|\hat{f}_{n,\tilde{h}_n}(x) - f(x)\|_\infty \leq \frac{f_{\min}}{2}$. It follows that $\sup_{x \in \mathbb{R}^d} \hat{f}_{n,\tilde{h}_n}(x) \leq f_{\max} + \frac{f_{\min}}{2}$, $\inf_{x \in \mathbb{R}^d} \hat{f}_{n,\tilde{h}_n}(x) \geq \frac{f_{\min}}{2}$, and

$$\left| \sum_{i \neq j} \mathbb{E}_X \left[ \tilde{\eta}_{n,h_n}^{(i)}(X) \, \tilde{\eta}_{n,h_n}^{(j)}(X) \right] - \frac{1}{n^2} \sum_{l,m} G_{lm,\tilde{h}_n} \theta_{lm} \right|$$

$$\leq \frac{1}{n^2} \sum_{l,m} K_{\tilde{h}_n}(\mathbf{x}_l - \mathbf{x}_m) \frac{\left| f^{\frac{1}{2}}(\mathbf{x}_l) f^{\frac{1}{2}}(\mathbf{x}_m) - \hat{f}_{n,\tilde{h}_n}^{\frac{1}{2}}(\mathbf{x}_l) \hat{f}_{n,\tilde{h}_n}^{\frac{1}{2}}(\mathbf{x}_m) \right|}{\hat{f}_{n,\tilde{h}_n}^{\frac{1}{2}}(\mathbf{x}_l) \hat{f}_{n,\tilde{h}_n}^{\frac{1}{2}}(\mathbf{x}_m) f^{\frac{1}{2}}(\mathbf{x}_l) f^{\frac{1}{2}}(\mathbf{x}_m)}$$

$$= \mathcal{O}\left( \sqrt{\frac{\log \tilde{h}_n^{-1}}{n\tilde{h}_n^d}} + \tilde{h}_n^\gamma \right) \cdot \frac{1}{n} \sum_{l=1}^{n} \hat{f}_{n,\tilde{h}_n}(\mathbf{x}_l)$$

$$= \mathcal{O}\left( \sqrt{\frac{\log \tilde{h}_n^{-1}}{n\tilde{h}_n^d}} + \tilde{h}_n^\gamma \right) = \mathcal{O}\left( \sqrt{\frac{\log h_n^{-1}}{nh_n^d}} + h_n^\gamma \right) \tag{23}$$

since $\tilde{h}_n = \sqrt{2}h_n$ Take $n_1 = \max\{n^{(1)}, n^{(2)}, n^{(3)}\}$, it follows from (20), (22) and (23) that with probability greater than $1 - 2QLh_n^{E_{\sigma_0^2}} - QLh_n^{E_{\sigma_1^2}}$,

$$R_n^{\text{PI}} = \frac{1}{n^2} \sum_{l,m} G_{lm,\sqrt{2}h_n} \theta_{lm} + \mathcal{O}\left( \sqrt{\frac{\log h_n^{-1}}{nh_n^d}} + h_n^\gamma \right) \tag{24}$$

and (5) is verified by (24). $\quad\square$

## 1.5  Proof of Corollary 1

Suppose the data $\{\mathbf{x}_i\}_{i=1}^{n}$ lies on a domain $\Omega \subseteq R^d$. Let $f$ be the probability density function on $\Omega$, $S$ be the cluster boundary which separates $\Omega$ into two parts $S_1$ and $S_2$ (see Figure 1). Let the domain of $f$ be restricted to $\Omega$ in assumption (A) and (B). Based on the analysis in the beginning of this document, Theorem $1-4$ and Lemma $1-2$ still hold and the proofs remain almost unchanged.

The Low Density Separation assumption favors the cluster boundary with low volume, i.e. $\int_S f(s) ds$. Corollary 1 reveals the relationship between the error of the plug-in classifier and the weighted volume of the cluster boundary.

Figure 1: Illustration of the hyperplane $S$ for Low Density Separation.

*Proof.* Firstly, we show that **when restricting the support of the marginal distribution $P_X$ to a subset $\Omega \subset \mathbb{R}^d$ which is not necessarily full-dimensional, Theorem 1 and lemma 1-3 still hold and our derived bounds are still valid.** To see this, we only need to show that the following class of functions $\mathcal{F}_\Omega$ is a bounded VC class of measurable functions.

$$\mathcal{F}_\Omega \triangleq \{K\left(\frac{t-\cdot}{h}\right), t \in \Omega, h \neq 0\} \tag{25}$$

Since we already know that the class of functions $\mathcal{F}$ defined below (also in the paper) is a bounded VC class of measurable functions with respect to the envelope function $F$,

$$\mathcal{F} \triangleq \{K\left(\frac{t-\cdot}{h}\right), t \in \mathbb{R}^d, h \neq 0\} \tag{26}$$

we have $N\left(\mathcal{F}, \|\cdot\|_{L_2(P)}, \tau \|F\|_{L_2(P)}\right) \leq \left(\frac{A}{\tau}\right)^v$ for every probability measure $P$ on $\mathbb{R}^d$ for which $\int F^2 dP < \infty$ and any $0 < \tau < 1$. $N\left(\mathcal{T}, \hat{d}, \epsilon\right)$ is defined as the minimal number of open $\hat{d}$-balls of radius $\epsilon$ required to cover $\mathcal{T}$ in the metric space $\left(\mathcal{T}, \hat{d}\right)$. Let $\{B_i\}$ be the $N\left(\mathcal{F}, \|\cdot\|_{L_2(P)}, \tau \|F\|_{L_2(P)}\right)$ open balls which cover $\mathcal{F}$, then $\{B_i \cap \Omega\}$ is the set of balls which cover $\mathcal{F}_\Omega$ since $\mathcal{F}_\Omega \subset \mathcal{F}$. It follows that $\mathcal{F}_\Omega$ is also a bounded VC class of measurable functions with respect to the envelope function $F$.

According to Theorem 3 in [2], for any $\varepsilon \in \left(0, \frac{1}{2}\right)$, there exists constant $C$ such that for all $h$ satisfying $0 < h < \sqrt{\tau}(2d)^{-\frac{e}{2(e-1)}}$,

$$\left| \frac{\sqrt{\pi}}{h} \int_{S_2} \int_{S_1} K_{\sqrt{2}h}(x-y)\, \psi_{\sqrt{2}h}(x)\, \psi_{\sqrt{2}h}(y)\, dxdy - \int_S f(s)\, ds \right| < Ch^{2\varepsilon} \tag{27}$$

where $\psi_h(x) = \frac{f(x)}{\sqrt{\int_\Omega K_h(x-z)f(z)dz}}$, $\tau$ is the radius of the largest ball that can be placed tangent to the manifold $\Omega$.

By the consistency of kernel density estimator in Lemma 2, there exists $n_0$ depending on $\sigma_0$ and the VC characteristics of $K$, when $n > n_0$, with probability greater than $1 - \tilde{L}h_n^{\frac{E_{\sigma_0^2}}{}}$, $\|\hat{f}_{n,\sqrt{2}h_n}(x) - f(x)\|_\infty \leq \frac{f_{\min}}{2}$. Define $R(\mathbf{x}_1, \ldots, \mathbf{x}_n) = \frac{\sqrt{\pi}}{2h_n} \hat{R}_n(\text{PI}_S) = \frac{1}{n^2} \frac{\sqrt{\pi}}{2h_n} \sum_{l,m} \frac{K_{\sqrt{2}h_n}(\mathbf{x}_l - \mathbf{x}_m)}{\hat{f}_{n,\sqrt{2}h_n}^{\frac{1}{2}}(\mathbf{x}_l) \hat{f}_{n,\sqrt{2}h_n}^{\frac{1}{2}}(\mathbf{x}_m)} \theta_{lm},$

then the bounded difference is verified:

$$\left| R(\mathbf{x}_1, \ldots, \mathbf{x}_l, \ldots, \mathbf{x}_n) - R(\mathbf{x}_1, \ldots, \mathbf{x}_l', \ldots, \mathbf{x}_n) \right|$$

$$\leq \frac{1}{n^2} \frac{\sqrt{\pi}}{h_n} \sum_{m \neq l} \left| \frac{K_{\sqrt{2}h_n}(\mathbf{x}_l - \mathbf{x}_m)}{\hat{f}_{n,\sqrt{2}h_n}^{\frac{1}{2}}(\mathbf{x}_l) \hat{f}_{n,\sqrt{2}h_n}^{\frac{1}{2}}(\mathbf{x}_m)} - \frac{K_{\sqrt{2}h_n}(\mathbf{x}_l' - \mathbf{x}_m)}{\hat{f}_{n,\sqrt{2}h_n}^{\frac{1}{2}}(\mathbf{x}_l') \hat{f}_{n,\sqrt{2}h_n}^{\frac{1}{2}}(\mathbf{x}_m)} \right|$$

$$= \frac{1}{n^2} \frac{\sqrt{\pi}}{h_n} \sum_{m \neq l} \frac{\left| \hat{f}_{n,\sqrt{2}h_n}^{\frac{1}{2}}(\mathbf{x}_l') K_{\sqrt{2}h_n}(\mathbf{x}_l - \mathbf{x}_m) - \hat{f}_{n,\sqrt{2}h_n}^{\frac{1}{2}}(\mathbf{x}_l) K_{\sqrt{2}h_n}(\mathbf{x}_l' - \mathbf{x}_m) \right|}{\hat{f}_{n,\sqrt{2}h_n}^{\frac{1}{2}}(\mathbf{x}_l') \hat{f}_{n,\sqrt{2}h_n}^{\frac{1}{2}}(\mathbf{x}_l) \hat{f}_{n,\sqrt{2}h_n}^{\frac{1}{2}}(\mathbf{x}_m)}$$

$$\leq \frac{C_1}{n h_n^{d+1}} \tag{28}$$

where $C_1$ a constant determined by $f_{\min}, f_{\max}, d$. According to McDiarmids Inequality,

$$\Pr\left[ |R(x_1, \ldots, x_n) - \mathbb{E}R(x_1, \ldots, x_n)| \geq \varepsilon_1 \right] \leq 2 \exp\left( -\frac{2 n h_n^{2d+2} \varepsilon_1^2}{C_1^2} \right) \tag{29}$$

and

$$\mathbb{E}R(x_1, \ldots, x_n)$$

$$= \frac{\sqrt{\pi}}{2h_n} \int_{S_2} \int_{S_1} \frac{K_{\sqrt{2}h_n}(x-y)}{\hat{f}_{n,\sqrt{2}h_n}^{\frac{1}{2}}(x) \hat{f}_{n,\sqrt{2}h_n}^{\frac{1}{2}}(y)} f(x) f(y) dx dy + \frac{\sqrt{\pi}}{2h_n} \int_{S_1} \int_{S_2} \frac{K_{\sqrt{2}h_n}(x-y)}{\hat{f}_{n,\sqrt{2}h_n}^{\frac{1}{2}}(x) \hat{f}_{n,\sqrt{2}h_n}^{\frac{1}{2}}(y)} f(x) f(y) dx dy$$

$$= \frac{\sqrt{\pi}}{h_n} \int_{S_2} \int_{S_1} \frac{K_{\sqrt{2}h_n}(x-y)}{\hat{f}_{n,\sqrt{2}h_n}^{\frac{1}{2}}(x) \hat{f}_{n,\sqrt{2}h_n}^{\frac{1}{2}}(y)} f(x) f(y) dx dy$$

Moreover, the square of the denominator of $\psi_h$ is the expectation of $\hat{f}_{n,h_n}$, i.e. $\int_\Omega K_h(x-z) f(z) dz = \mathbb{E}\left[ \hat{f}_{n,h} \right]$. By equation $(a)$ in the proof of Lemma 2,

$$\Pr\left[ \| \int_{S_2} \int_{S_1} \frac{K_{\sqrt{2}h_n}(x-y)}{\hat{f}_{n,\sqrt{2}h_n}^{\frac{1}{2}}(x) \hat{f}_{n,\sqrt{2}h_n}^{\frac{1}{2}}(y)} f(x) f(y) dx dy - \int_{S_2} \int_{S_1} K_{\sqrt{2}h_n}(x-y) \psi_{\sqrt{2}h_n}(x) \psi_{\sqrt{2}h_n}(y) dx dy \|_\infty \geq \varepsilon_2 \right]$$

$$\leq \Pr\left[ \| \mathbb{E}\left[ \hat{f}_{n,\sqrt{2}h_n}(x) \right] - \hat{f}_{n,\sqrt{2}h_n}(x) \|_\infty \geq C_2 \varepsilon_2 \right]$$

$$\leq L \exp\left( -\frac{1}{L} \frac{\log(1 + \lambda/4L)}{\lambda} \frac{n(\sqrt{2}h_n)^d C_2^2 \varepsilon_2^2}{\sigma_0^2} \right) \tag{30}$$

where $C_2$ is a constant. By (27), (29) and (30) and the application of the Borel-Cantelli lemma, (7) is verified. $\qquad\square$

## 1.6 Proof of Theorem 2

*Proof.* Denote the support of $P_X$ by $\mathcal{X}$. Since $\mathcal{X}$ is bounded in $\mathbb{R}^d$, we construct the $\tau$-cover of $\mathcal{X}$ which is a sequence of sets $\{\Omega_1, \Omega_2, ..., \Omega_\mathcal{R}\}$ such that $\mathcal{X} \subseteq \bigcup_{r=1}^{\mathcal{R}} \Omega_r$ and each $\Omega_r$ is a box of length $\tau$ in $\mathbb{R}^d$, $1 \leq r \leq \mathcal{R}$, $\mathcal{R} = \left( \frac{2M_0}{\tau} \right)^d$. Let $A = \bigcap_{r=1}^{\mathcal{R}} \{\Omega_r \bigcap \{\mathbf{x}_l\}_{l=1}^n \neq \emptyset\}$ indicating the event that each $\Omega_r$ contains at least one data point from $\{\mathbf{x}_l\}_{l=1}^n$, then

$$\Pr[A] \geq 1 - \mathcal{R}(1 - \Pr[\Omega_1])^n = 1 - \mathcal{R}e^{n \log(1 - \Pr[\Omega_1])}$$

$$\geq 1 - \mathcal{R}e^{-n \Pr[\Omega_1]} \geq 1 - \left( \frac{2M_0}{\tau} \right)^d e^{-n f_{\min} \tau^d}$$

So that $A$ holds with probability greater than $1 - \left( \frac{2M_0}{\tau} \right)^d e^{-n f_{\min} \tau^d}$. Denote by $\tilde{X}$ the nearest neighbor of $X$ among $\{\mathbf{x}_l\}_{l=1}^n$, and $\tilde{Y}$ is the label of $\tilde{X}$. Note that $\left\| X - \tilde{X} \right\|_2 \leq \sqrt{d}\tau$ if $X \in \Omega_r$

for each $r$. For any $P_{XY} \in \mathcal{P}_{XY}$, some calculation shows that $\exists \tilde{c}_i > 0, \left| \eta^{(i)}(x) - \eta^{(i)}(y) \right| \leq \tilde{c}_i \|x - y\|^\gamma$, so that $\eta^{(i)}$ is also Hölder-$\gamma$ smooth with Hölder constant $\tilde{c}_i$. We then have

$$R\left(\text{NN}_S\right) = \mathbb{E}_{(X,Y)}\left[Y \neq \tilde{Y}\right] \tag{31}$$

$$= \sum_{r=1}^{\mathcal{R}} \mathbb{E}_X\left[\left(1 - \eta^{(\tilde{Y})}(X)\right) \mathbb{1}_{\{X \in \Omega_r\}}\right]$$

$$\leq \sum_{r=1}^{\mathcal{R}} \mathbb{E}_X\left[\left(1 - \eta^{(\tilde{Y})}(\tilde{X}) + \tilde{c}_{\tilde{Y}}\left(\sqrt{d}\tau\right)^\gamma\right) \mathbb{1}_{\{X \in \Omega_r\}}\right]$$

$$\leq \sum_{r=1}^{\mathcal{R}} \mathbb{E}_X\left[\left(1 - \eta^{(\tilde{Y})}(\tilde{X})\right) \mathbb{1}_{\{X \in \Omega_r\}}\right] + \underbrace{\max_i \tilde{c}_i}_{\triangleq c_0} \left(\sqrt{d}\tau\right)^\gamma$$

Let $N_r = \{\mathbf{x}_s \in \{\mathbf{x}_l\}_{l=1}^n \mid \mathbf{x}_s = \tilde{X} \text{ for some } X \in \Omega_r\}$ wherein each element is the nearest neighbor of some $X \in \Omega_r$, and $\Omega_{rs} = \{X \in \Omega_r \mid \tilde{X} = \mathbf{x}_s, \mathbf{x}_s \in N_r\}$ which is a subregion of $\Omega_r$ such that all $X \in \Omega_{rs}$ takes $\mathbf{x}_s$ as its nearest neighbor. Then $\Omega_r = \bigcup_{s:\mathbf{x}_s \in N_r} \Omega_{rs}$, and $\tilde{X} = \mathbf{x}_s$ for $X \in \Omega_{rs}$. Since $\{\mathbf{x}_l\}_{l=1}^n \subset \bigcup_{r=1}^{\mathcal{R}} \Omega_r$, each $\mathbf{x}_l$ should be the nearest neighbor of some $X \in \Omega_r$, $1 \leq r \leq \mathcal{R}$, so that $\{\mathbf{x}_l\}_{l=1}^n = \bigcup_{r=1}^{\mathcal{R}} N_r$.

Based on Theorem 1, with probability greater than $1 - 2QLh_n^{\frac{E_{\sigma_0^2}}{}}$,

$$\sum_{r=1}^{\mathcal{R}} \mathbb{E}_X\left[\left(1 - \eta^{(\tilde{Y})}(\tilde{X})\right) \mathbb{1}_{\{X \in \Omega_r\}}\right]$$

$$= \sum_{s=1}^{n} \left[1 - \eta^{(y_s)}(\mathbf{x}_s)\right] \int_{\mathcal{V}_s} f(x)\,\mathrm{d}x$$

$$= \sum_{s=1}^{n} \left\{\left[1 - \hat{\eta}_{n,h_n}^{(y_s)}(\mathbf{x}_s)\right] \int_{\mathcal{V}_s} \hat{f}_{n,h_n}(x)\,\mathrm{d}x\right\} + \mathcal{O}\left(\sqrt{\frac{\log h_n^{-1}}{nh_n^d}} + h_n^\gamma\right)$$

$$= \frac{1}{n} \sum_{l<m} H_{lm}\theta_{lm} + \mathcal{O}\left(\sqrt{\frac{\log h_n^{-1}}{nh_n^d}} + h_n^\gamma\right) \tag{32}$$

where $\mathcal{V}_s$ is the Voronoi cell associated with $\mathbf{x}_s$, which is the set of points whose nearest neighbor is $\mathbf{x}_s$: $\mathcal{V}_s = \bigcap_{l:l \neq s} \{x \in \mathcal{X} \mid \|x - \mathbf{x}_s\|_2 \leq \|x - \mathbf{x}_l\|_2\}$. Combining (31) and (32),

$$R\left(\text{NN}_S\right) \leq \frac{1}{n} \sum_{l<m} H_{lm}\theta_{lm} + c_0\left(\sqrt{d}\tau\right)^\gamma + \mathcal{O}\left(\sqrt{\frac{\log h_n^{-1}}{nh_n^d}} + h_n^\gamma\right) \tag{33}$$

Moreover, the equality in (33) holds if the equality in (31) holds, e.g. $\eta^{(i)} \equiv \frac{1}{Q}$ for $1 \leq i \leq Q$. $\square$

## 2 Algorithm and Experiments

The objective function of our pairwise clustering method PIEC is below:

$$\Psi(e) = \sum_{l=1}^{n} \exp\left(-G_{le_l, \sqrt{2}h_n}\right) + \lambda \sum_{l,m} \left(\tilde{\theta}_{lm} G_{lm, \sqrt{2}h_n} + \rho_{lm}(e_l, e_m)\right) \tag{34}$$

where $\rho_{lm}$ is a function to enforce the consistency of the cluster indicators:

$$\rho_{lm}(e_l, e_m) = \begin{cases} \infty & e_m = l, e_l \neq l \text{ or } e_l = m, e_m \neq m \\ 0 & \text{otherwise} \end{cases},$$

The minimization of the objective function is converted to a MAP (Maximum a Posterior) problem in the pairwise MRF. (34) is minimized by Max-Product Belief Propagation (BP) in two steps:

**Message Passing:** BP iteratively passes messages along each edge according to

$$m_{lm}^t(e_m) = \min_{e_l} \left( M_{lm}^{t-1}(e_l) + \tilde{\theta}_{lm} G_{lm, \sqrt{2} h_n} + \rho_{lm}(e_l, e_m) \right) \tag{35}$$

$$M_{lm}^t(e_l) \triangleq \sum_{k \in \mathcal{N}(l) \backslash m} m_{kl}^t(e_l) + u_l(e_l) \tag{36}$$

where $m_{lm}^t$ is the message sent from node $l$ to node $m$ in iteration $t$, $\mathcal{N}(l)$ is the set of neighbors of node $l$.

**Inferring the optimal label:** After the message passing converges or the maximal number of iterations is achieved, the final belief for each node is $b_l(e_l) = \sum_{k \in \mathcal{N}(l)} m_{kl}^T(e_l) + u_l(e_l)$, $T$ is the number of iterations of message passing. The resultant optimal $e_l^*$ is $e_l^* = \arg\min_{e_l} b_l(e_l)$.

AP (Affinity Propagation) controls the cluster numbers by a parameter called preference. We first estimate the lower bound and upper bound for the preference using the routine functions provided by the authors [3], then evenly sample 170 preference values between its upper bound and lower bound, and run AP with each sampled preference value. CEB (Convex Clustering with Exemplar-Based Model) produces different cluster numbers by varying the scale $\beta \beta_0$ which controls the shape of the mixture components. Likewise, we evenly sample 170 values between $[0.1, 2]$ for $\beta$, and $\beta_0 = n^2 \log n / \sum_{i,j} \|\mathbf{x}_i - \mathbf{x}_j\|_2^2$ according to [4]. Also, we normalize the BT data set so that it has unit column variance, since the column variances of BT vary significantly (the largest column variance is 18580 while the smallest one is 0.0686).