[Reviews · NeurIPS 2014]

Submitted by Assigned_Reviewer_11

This paper proposes a new pairwise clustering framework where nonparametric pairwise similarity is derived by minimizing the generalization error unsupervised nonparametric classifier. The proposed framework bridges the gap between clustering and multi-class classification, and explains the widely used kernel similarity for clustering. The authors also prove that the generalization error bound for the unsupervised plug-in classifier is asymptotically equal to the weighted volume of cluster boundary for low density separation. Based on the derived nonparametric pairwise similarity using the plug-in classifier, the authors propose a new nonparametric exemplar-based clustering method with enhanced discriminative capability compared to the exiting exemplar-based clustering methods.

Here are few comments.

1. In equation (6) which which provides and estimate for the regression function, shouldn’t there be a \pi^{(i)} in the numerator as was in line 123?

2. In lemma 2, kernel bandwidth h should be h_n

3. typo: line 63, remove double the.

3. The theoretical results are interesting and nice, especially the connection to the low density separation is interesting.

4. While the theoretical results are nice, its application towards developing new nonparametric examplar based clustering algorithm looks a little bit complex.

Overall, this is an interesting paper.
Summary: Overall, this is an interesting paper.

Submitted by Assigned_Reviewer_13

This paper presents clustering framework using pairwise similarity function model.
Different data partitions are used to construct different nonparametric classifiers.
The optimal clustering is obtained by minimizing the generalization error of the
learned classifiers related to data partitions.
The authors consider two nonparametric classifiers in their framework:
the plug-in and the nearest-neighbor classifier.
The authors prove also a result interesting in itself, namely that the generalization
error bound for the unsupervised plug-in clustering is asymptotically equal to the
weighted volume of cluster boundary for Low Density Separation.

This paper tights together two important machine learning subjects: multi-class
classification problem and clustering. Furthermore, it has strong theoretical background.
The paper is however very poorely written.
The introduction is well-organized however in the next sections the authors
go into several technical details and do not summarize their results in the
compact understandable form. Two technical lemmas given in Section 2 only deepen
the confusion. Thus, it is extremely difficult to measure the real impact of this paper.
Section 2 contains too many technical details that are given before giving the reader any sort of intuition.
Besides the presented lemma statements are completely unclear.
For instance:

Lemma 1:

- is it possible to say how n_{0} depends on \sigma_{0} and VC characteristics of K ?
- the statements seems not to be put in the grammatical form, the authors should review it

Lemma 2:

- again the authors introduce many parameters that mutually depend on each other,
it is almost impossible to guess from all the asusmptions required by the lemma what it
says in fact

Lemma 3:

- similar problems to those mentioned above

Lemma statements are unreadable and some of them are put at the very beginning of the paper.
The authors should spend much more time to work on the presentation of their results.
Maybe some summary of the obtained theoretical results should be conducted before giving exact statements.
The authors should also give some intuition regarding all lemmas presented in the main body of the paper.

The authors present how their techniques may be applied to the exemplar-based clustering but the comparison
with existing state-of-the-art methods is missing. The experimental section is very tiny and does not shed
any light of advantages of the technique developed by the authors over other clustering algorithms.
The difficulty of the clustering heavily depends on the considered setting: whether data is truly high-dimensional,
or maybe lies on the small-dimensional manifolds, whether data points are sparse or dense,
whether the groundtruth clusters are convex or there are no
assumptions about their shape, what is the objective function the authors aim to minimize or the groundtruth
clustering they want to approximate well. Without detailed analysis of the performance of the algorithm in the specific
setting defined by these parameters it is almost impossible to say whether the presented method can be applied in
practice and solves a nontrivial problem.

What is the computational complexity of the algorithms that use presented technique ? This issue is not addressed at all.
Is it possible to extend the analysis for other classifiers (only nearest-neighbor and plug-in classifiers were analyzed) ?
Is it possible to release some source code files (with the implementation of the presented method) that were used by
the authors to test their approach ? (The last remark is not crucial though.)
Summary: To sum it up, this paper focuses on the very important machine learning problem and is somehow innovative (presents
new ideas that according to what authors say: "bridge the gap between clustering and the multilabel classification"; to
some extend I do agree with aurhors' statement). However it seems to me that the authors do not spend much time on
preparing a readable version of the paper. The presentation is chaotic, it is very hard to understand what are the main
theoretical results of the paper. The experimental section is incomplete. No comparison with currently existing methods is conducted.
I would suggetst the authors to spend much more time on completely reorganizing the paper according to the points I mentioned above.
In particular, sections: 2 and 3 should be completely rewritten.

Submitted by Assigned_Reviewer_34

In this paper, the authors take a different look at discriminative clustering. Their central idea is the following: they imagine a classifier that separates the different (cluster) classes, and derive a bound for the generalization error induced by this classifier. This is the bound that they then attempt to optimize using a clustering algorithm (in this case, a belief-propagation based method).

The main feature that distinguishes this work from prior work is that they reformulate the cost function as a new kind of similarity measure that makes use of a kernel density estimate, so as to avoid a parameter search problem. This is not to say that they don't have parameters in their formulation: but these are either balance paramters or a variance parameter for the kernel density function they estimate.

Overall, I think this is a reasonable idea, and does have some merit in terms of reducing the number of parameters for discriminative clustering. It also provides a better story about what exactly the clustering algorithm is attempting to optimize.

In terms of whether the idea has merit in practice, the paper is a little thinner. The main comparison is to a different exemplar-based clustering approach. This to me seems a little odd, since one of the claimed selling points of the paper is that it requires fewer parameters than other discriminative clustering methods. Would it have been more useful to compare to MM clustering or even the information-theoretic approaches as well ? The space spent proving consistency of the KDE could very well have been utilized for such a comparison, because the consistency results follow fairly directly from known results.

I didn't find the paper very easy to read, mainly because of the excess of notation used to explain the main ideas. It would have benefited (somewhere) from a final "here is our algorithm" explanation that puts everything together. While I understand how the pieces fit together, I spent some time trying to track down various parameters (including the variance h) and how they were being used in the overall algorithm.
Summary: A reasonable idea that deserves consideration. But could do with a more thorough experimental evaluation.
Author Feedback
Author rebuttal: We appreciate the comments from the reviewers. All the reviewers unanimously agree that this paper has solid theoretical background and it bridges the gap between multi-class classification and clustering from a novel angle. We will focus mainly on the presentation and the experiment issue raised by reviewer_13. The references are cited in the same way as in the paper.

To reviewer_11:
Thanks for your positive review.

To reviewer_13:
Lemma 1 and Lemma 2 state the strong consistency (almost sure uniformly convergence) of the traditional kernel density estimator and the generalized kernel density estimator. These results of strong consistency are used as the primary tools for the generalization error bounds for the classification models (Theorem 1 and Theorem 2), and such generalization error bounds are the main theoretical contribution of this paper.
We have carefully scrutinized the statements of Lemma 1-3, and we insist that the statements are readable. There are no mutually dependent parameters in Lemma 2 and Lemma 3, and all the assumptions are clear in the lemma statements. n_{0} can be expressed as a function of \sigma_{0} and L, C (i.e. the VC characteristic of the kernel K ) according to Corollary 2.2 in [14]. Note that the functional form of n_{0} can further complicate the statements of the lemmas, so we do not introduce the explicit form of n_{0}. It is important to note that the statistics literature also adopt this style, i.e. below is a representative paper which also uses the results of Corollary 2.2 in [14]:

Rinaldo, Alessandro; Wasserman, Larry. Generalized density clustering. The Annals of Statistics 38 (2010), no. 5, 2678—2722.

In propagation 9 of the above paper, there is a similar n_{0} and the paper explicitly states that n_{0} depends upon several other variables.
In our revised draft, we do present more intuitions and summary of the theoretical results before stating our lemmas the theorems in Section 2 and Section 3, which is available at the anonymous website:

https://sites.google.com/site/nips14paper126/

It should be emphasized that the major contribution of our paper is the novel pairwise clustering scheme which unifies classification and clustering, and the related theoretical results. Limited by the space, our experiment part cannot elaborate every aspect of our practical algorithm. We demonstrate the advantage of our algorithm in the context of nonparametric exemplar-based clustering, and we do compare our results to other state of the art exemplar-based clustering method in our paper. To the best of our knowledge, all the exemplar-based clustering methods use some proper pairwise similarity measure to obtain the underlying clusters in the data and identify the corresponding cluster exemplars. For example, Affinity Propagation (AP) uses the user-given pairwise similarity (usually the Euclidean distance), and Convex Clustering with Exemplar-Based Model (CEB) uses the Mahalanobis distance in a mixture model. With the nonparametric pairwise similarity derived in our framework, our Plug-In Exemplar Clustering method achieves better results than the current exemplar-based clustering methods, which demonstrates the advantage of our pairwise similarity measure. We provide more comparison results at the aforementioned website.

The computational complexity of our clustering algorithm is O(TEN), where E is the number of edges in the pairwise MRF, T is the number of iterations of message passing in Belief Propagation algorithm, and n is the number of data points. Extending the analysis for other classifiers is possible, and we treat it as our future work.

To reviewer_34:
Thanks for your positive review on our paper. Please access the anonymous website https://sites.google.com/site/nips14paper126/ to view more comparison results. We will compare our method to MM clustering and information theoretic methods in our future work, where we will focus on the empirical evaluation of our clustering algorithm.